# Four Virtual-Reality Simulators for Diagnostic Abdominal Ultrasound Training in Radiology

**DOI:** 10.3390/diagnostics9020050

**Published:** 2019-05-06

**Authors:** Mia Louise Østergaard, Lars Konge, Niklas Kahr, Elisabeth Albrecht-Beste, Michael Bachmann Nielsen, Kristina Rue Nielsen

**Affiliations:** 1Department of Radiology, North Sealand Hospital, 3400 Hillerød, Denmark; 2Copenhagen Academy for Medical Education and Simulation, 2100 Copenhagen, Denmark; lars.konge@regionh.dk; 3Department of Radiology, Rigshospitalet, 2100 Copenhagen, Denmark; niklaskahr@gmail.com (N.K.); mbn@dadlnet.dk (M.B.N.); marie.kristina.rue.nielsen.01@regionh.dk (K.R.N.); 4Department of Clinical Physiology, Nuclear Medicine and PET, Rigshospitalet, 2100 Copenhagen, Denmark; elisabeth.albrecht-beste@regionh.dk

**Keywords:** simulation, ultrasound, education, abdominal, virtual-reality, radiology

## Abstract

Ultrasound exams need skilled examiners, and simulation-based training could provide standardized and safe skill training. This study aims to give an overview of different types of virtual-reality simulators for use in abdominal diagnostic ultrasound training in radiology. Fifteen specialized radiologists and radiological residents were presented with two similar cases on four different simulators for abdominal ultrasound training. A feedback sheet for each individual simulator and for an overall comparison was filled out by all participants. All means of scores were compared, and simulators were ranked from least to most favorable. One simulator was ranked most favorable in seven out of nine questions, but none of the mean scores had statistically significant differences. All simulators were recommended for training radiologists, and all simulators were perceived to benefit trainees more than experienced ultra-sonographers.

## 1. Introduction

Ultrasound is a safe and readily available diagnostic tool that, in the hands of a skilled examiner, can play a crucial role in the diagnostic process. In radiology, the diagnostic abdominal ultrasound exam is a highly valued skill in the practical curriculum [1]. Supervised scanning on patients can provide these skills, but relies on patient load, randomly represented pathologies, and the availability of expert supervisors. New virtual-reality simulators provide standardized training conditions, systematic case presentations including rare but crucial pathologies, and readily available automatic feedback [2].

Several simulators are available for training the diagnostic abdominal ultrasound examination using different types of technics and interfaces. Each have their own strengths and weaknesses, and all come at a certain cost and need trained instructors to oversee the training. Training on simulators can be effective and even cost-effective when the right simulator is chosen, but no current studies have explored which simulator best fits a specific need [3,4]. This study aims to give an overview of different types of virtual-reality simulators for use in abdominal diagnostic ultrasound training in radiology.

## 2. Materials and Methods

The participants in this study were specialized radiologists with more than five years of ultrasound experience or physicians starting specialization in radiology with less than six months of ultrasound experience (Table 1). They were recruited from two hospitals and one private ultrasound practice in Copenhagen, Denmark. Data collection was done in a standardized setting at Copenhagen Academy for Medical Education and Simulation, Rigshospitalet in June 2018 [5].

By searching the internet, five simulation companies were identified as having abdominal ultrasound simulators for diagnostic use. One company declined to participate, and four simulators were assigned a letter in alphabetical order and evaluated: A = ScanTrainer (Intelligent Ultrasound), B = Schallware, C = Simbionix (3D Systems), and D = Vimedix (Figure 1, Table 2).

All participants were given a short, standardized introduction to each simulator, including identification of gain, debt and focus buttons, feedback options, and probe handling. Each simulator was presented with two cases: a patient with a hyperechoic liver tumor and a patient with free fluid visible in the splenorenal space. Participants were not presented with the more detailed information on each simulator, e.g., price, number of cases, or other available modules. The participants evaluated the simulators in a randomized order to minimize the risk of bias.

Each participant had a maximum of three minutes to view each case, and, after completing both cases on each simulator, a standardized feedback sheet was filled out for that particular simulator. The feedback sheets were identical for all simulators, and participants were asked to evaluate three areas of educational value: fidelity, usability, and overall satisfaction respectively, on a five-point Likert’s scale, as shown in Table 3. After all four simulators were individually evaluated, an overall questionnaire was filled out ranking the simulators from most to least favorable in nine different categories. Free text comments were possible on all questionnaires and all comments are included in the results.

The means of scores for the overall questionnaires were compared for each question, and the first, second, third, and fourth places were represented by green, orange, yellow, or red markers respectably, as seen in Table 3.

The mean scores for the individual and overall questionnaires were compared between the experienced radiologists and the residents to explore potential different perceptions between experts and trainees.

## 3. Results

A total of 15 participants completed the simulator evaluation: eight experienced radiologists and seven radiological residents. 

Combined overall ranking was most favorable for Schallware (simulator B) in seven of the nine ranked areas (resembles real scanning images, probe handling, knobology, case description, hints, and overall feel). The experienced radiologists gave the same simulator best ranking in all categories, while the residents gave it best ranking in two categories (resembles real scans and knobology) and worst in one (probe handling). Best ranking from the residents was given to 3D systems (simulator C) in six of the nine categories (educational value, probe handling, intuitive use, hints, feedback, and overall feel), while one worst rankings was given by the experienced radiologists (resembles real scans). Intelligent Ultrasound (simulator A) was ranked second best in two categories (resembles real scans and case description) in the combined score and best by the residents in case description. The combined ranking put CAE Healthcare (simulator D) as third best in four categories (educational value, resembles real scans, intuitive use, and hints), while residents ranked the same simulator second-best in probe handling and intuitive use.

As seen in Table 4, there were no statistically significant differences between the perceptions of experienced ultrasonographers and trainees in any of the four main topics: educational value, fidelity, usability, and overall satisfaction. As seen in Table 5, the collective scores for educational value were similar between the four simulators, with all simulators scoring higher for educational use for novices than for intermediates and receiving the lowest scores for educating advanced learners. Fidelity scores indicated that Simulator B had the best images. Probe handling was judged positively for all simulators using mannequins (B, C, and D), while knobs were rated high for all simulators not using a mouse (A, B, and C). All simulators scored high for intuitive use and slightly lower for providing useful instructions and feedback. An above-average score was awarded to all simulators for easy use, and for the overall feel for two of the simulators (B and C). Simulator B was specifically recommended for radiological departments, and no simulators were considered non-recommendable, as seen in Table 5. 

The most commented-on item with eight mentions for all simulators was the wish for keyboard/knobs resembling a clinical ultrasound machine, as none of the simulators had turning or click buttons, as seen on most conventional scanners. Simulator A was thought to have distorted images by one reviewer, and another had trouble without a physical torso to scan on. Two participants found the lack of full body scans for all cases problematic for simulator B, while two participants commented that the scans were highly realistic. Simulator C was thought to have unrealistic images by four participants, and three found the anatomical markings on the image very helpful and learner-friendly. For simulator D, the images were also found to be unrealistic by three participants; one commented that the depth was too automated, two thought it highly suitable for beginners, and one resident simply “loved it”.

Overall comments stated that the simulated images were generally easier to obtain than real-life images and novices therefore could get a head start before scanning on patients. The use of mannequins was generally favored. The feel of the shirt-covered mannequin for simulator B was positively commented on by two participants as being close to the feeling of a probe on real skin with scanning gel, and one participant preferred the feel of simulator D’s mannequin. Two participants wished for the images from simulator B with the anatomical help markers of simulator C or D.

## 4. Discussion

This study compared four virtual-reality abdominal ultrasound simulators for diagnostic training in radiology. One simulator (B) was positively rated as recommendable, and none of the simulators were rated as non-recommendable. 

Simulation training is being increasingly implemented in the curriculum for specialized training of physicians, and ultrasound training has a high-ranking placement in the national needs assessment for radiology in Denmark [1,6].

All simulators can be used to provide basic skills training outside the clinic without burdening the patients, as well as provide a standardized and uniform training platform in a safe environment [7]. All simulators in this study have been used for educational development, and some have been used to produce training set ups and tests with validity evidence. Training on the CAE Vimedix showed improved examination scores on visual skills for 10 medical students after simulation training, and a study with 56 medical students training on the Intelligent Ultrasound supported the use of errors as a training tool for better transfer of skills to the clinical environment [8,9]. A study with 60 medical students also using the Intelligent Ultrasound simulator supported the use of individualized training on simulators to steepen the learning curve for trainees. The same simulator was used to show that performance of FAST (focused assessment with sonography for trauma) scans could be assessed in a non-clinical setting on simulators [10,11]. The Schallware simulator was used for test development for the diagnostic abdominal ultrasound exam using the European curriculum, and training on the simulator prior to clinical training was shown to improve scanning performance on patients [12,13]. The Schallware simulator was also used to establish a valid assessment tool for FAST and shown to provide basic skills for physicians using FAST [14].

The use of simulators could relieve some of the pressure on the clinical patients and supervisors as well as provide a safe and standardized educational environment for trainees [2]. This could provide a more unified knowledge basis of trainees and ensure that all trained personnel have a basic skill level. Optimized learning strategies in an environment with increasing safety and efficiency demands are also possible with the use of simulators [15]. This could include goal setting, continued skill assessments, spacing practice sessions, and more. Finally, physicians’ skill development would be ensured while providing standardized skill assessment and curriculums [16]. The use of simulation training has been shown to be effective for skill development in many medical specialties, e.g., gynecology and respiratory medicine, and a randomized controlled trial from 2015 found simulation training to be more effective than the apprenticeship model [3,17].

All simulators in this study could provide several pathological scans for image recognition training and a broad variety of scans such as FAST, POC (point of care), and TEE (transesophageal echocardiogram). 

### Limitations

All simulators were evaluated by radiologists who are used to high-end scanning equipment and interpreting a broad array of image details. Therefore, the ranking might not reflect use for training more constricted diagnostic tasks such as FAST or eFAST (extended focused assessment with sonography for trauma). The limited number of participants also calls for care of interpretation and elaboration of the results. As simulators are not the same as real life, care should be taken to find other ways to train the non-simulated areas of diagnostic ultrasound such as doctor-patient communication and pressure sensitivity. However, simulation training in abdominal ultrasound has been shown to elevate the trainee before scanning on patients with possible benefits for patients and supervisors as well as the trainee [18]. The Doppler features of the simulators were not presented due to time restrictions, and are therefore not included in the ranking score. The field of simulation is moving fast, and new simulators are emerging rapidly, e.g., Intelligent Ultrasound launched a mannequin-based simulator shortly after this study ended, and 3D Systems has changed their name from Simbionix to 3D Systems.

## 5. Conclusions

All simulators were usable for diagnostic abdominal ultrasound, and learning benefits were ranked higher for novices than for more experienced ultrasonographers. Hopefully, this study can assist decision makers in ultrasound training programs to choose the simulator that optimally fits their local needs and budget.

## Figures and Tables

**Figure 1 diagnostics-09-00050-f001:**
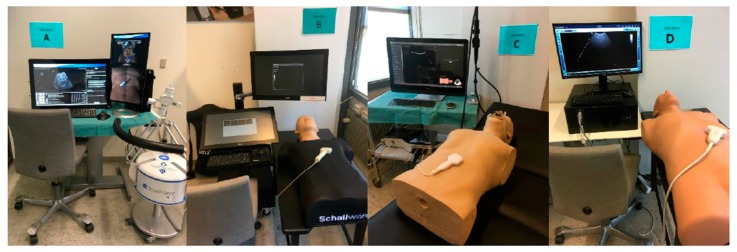
Simulators in the Study’s Setting.

**Table 1 diagnostics-09-00050-t001:** Demographics.

Demographics	Radiological Residents	Radiologists	All
**Group size**	7	8	15
Female	3	2	5
Male	4	6	10
**Ultrasound experience in weeks (SD)**	13.9 (17.5)	1093 (491.7)	-

**Table 2 diagnostics-09-00050-t002:** The Simulators.

Simulator	MedaPhor	Schallware	Simbionix (3D Systems)	CAE Healthcare
**Study mark**	A	B	C	D
**Version**	ScanTrainer TVS & TAS, 2017x	Ultrasound Simulator Station-128	U/S Mentor	CAE Vimedix
**Primary based**	Cardiff, UK Available worldwide	Berlin, Germany Available worldwide	Tel Aviv, Israel Available worldwide	Sarasota, Florida, US Available worldwide
**Description**	A hard drive, two screens, a floor-mounted haptic device, keyboard and mouse with rollerball. One screen displays image, buttons and help features, and the other screen displays the virtual patient and probe	A hard drive, two touch screens, keyboard with a rollerball, probes and a sensor table with a mannequin torso. One screen mimics ultrasound buttons and one shows the scan image	A hard drive, one touch screen with screen divisions, keyboard, probes and a sensor table with a mannequin torso. The split screen displays the scan image on one side, and buttons plus anatomical helping model on the other side	A hard drive, one split-screen, keyboard, mouse, probes and a sensor table with a mannequin torso. The screen displays the scan image, buttons and help features
**Price**	€20,000 to €90,000	€20,000 to €65,000	50,0000 to 100,000 USD	Not disclosed
**Abdominal cases**	About 250	About 250	8 diagnostic and 11 FAST	10 cases
**Custom cases**	Yes, customers can upload their own scans from patients	Yes, with additional equipment	All cases can be customized for severity and/or presence of pathology	All cases can be customized for severity and/or presence of pathology
**Source of scan**	Real patient CT scans blended with computer data	Real patient ultrasound b-mode scans	Computer generated data	Computer generated data
**Feedback options**	ScanTutor will test user against set metrics. Metrics can be customized. Full diagnostic list for each case	Region of interest (RoI) with the option of turning RoIs into questions/answers. Full diagnostic list for each case	Case severity feature/multiple scenarios pr. case. Skill tasks. Full diagnostic list for each case	Performance assessments with kinematic metrics. Full diagnostic list for each case
**Dynamic features**	Computer animated doppler incl. CFM, PW & CW. Breathing patient	Doppler pre-recorded	Computer animated Doppler Breathing patient	Computer animated Doppler Breathing patient
**Other modules available**	OBGYN eFAST/FAST Emergency Medicine General Medicine Breast	Obstetrics/Gynecology Cardiology TEE Pediatric (new born manikin) Irritable Boule Disease Virtual model, adult Heart/lungs animated Virtual fetus model	Basic sonography skills Obstetrics/Gynecology Fetal Neurosonography + Echo Cardiology TEE/Echo eFAST/RUSH Emergency Medicine Lung	Obstetrics/Gynecology Cardiology TEE FAST/eFAST Emergency Medicine

Simulator information, obtained from intelligentultrasoundsimulation.com, schallware.de, simbionix.com, and cae.com.

**Table 3 diagnostics-09-00050-t003:** Comparative Questionnaires.

Overall Ranking	MedaPhor	Schallware	Simbionix	ViMedix
**Overall Educational value**	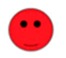	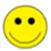	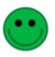	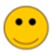
Residents/Experienced	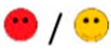	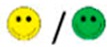	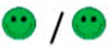	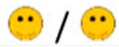
**Resembles real scanning on image**	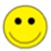	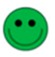	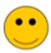	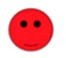
Residents/Experienced.tif	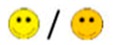	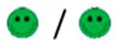	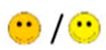	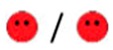
**Resembles real scanning on probe handling**	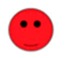	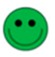	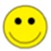	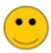
Residents/Experienced	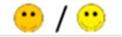	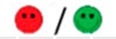	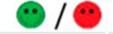	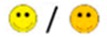
**Resembles real scanning knobology**	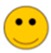	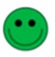	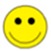	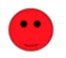
Residents/Experienced	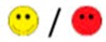	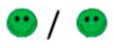	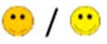	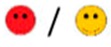
**intuitive Use**	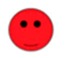	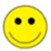	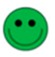	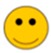
Residents/Experienced.tif	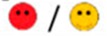	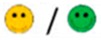	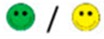	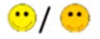
**Case description**.	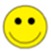	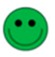	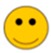	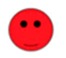
Residents/Experienced	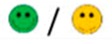	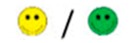	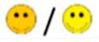	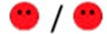
**Usable hints**	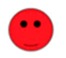	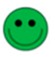	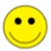	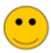
Residents/Experienced	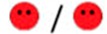	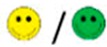	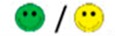	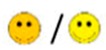
**Useful Feedback**	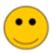	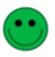	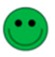	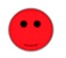
Residents/Experienced	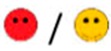	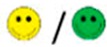	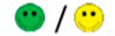	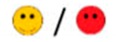
**Overall feel**	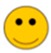	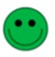	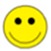	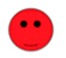
Residents/Experienced	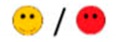	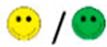	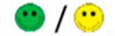	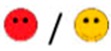

Overall ranking with first, second, third, and fourth place, represented by green, yellow, orange, and red smileys, respectively. The collective ranking is shown (large smiley) with the ranking divided by groups underneath (small smileys). If ranking scores are identical, two simulators share the respective placing, e.g., two simulators share first place in feedback, and no second placing is listed.

**Table 4 diagnostics-09-00050-t004:** Comparing Mean Scores by Groups.

Comparing Mean Score by Groups	Radiological Residents	Radiologists	*P* Value
**Simulator A**			
Educational Value (SD)	3.1 (1.2)	2.9 (1.2)	>0.05
Fidelity (SD)	2.9 (0.9)	3.3 (1.3)	>0.05
Usability (SD)	3.1 (0.9)	3.5 (1.3)	>0.05
Overall Satisfaction (SD)	2.9 (1.4)	2.9 (1.6)	>0.05
**Simulator B**			
Educational Value (SD)	4.0 (0.6)	3.8 (0.8)	>0.05
Fidelity (SD)	3.8 (0.8)	4.1 (0.5)	>0.05
Usability (SD)	4.1 (0.8)	3.9 (0.9)	>0.05
Overall Satisfaction (SD)	3.8 (1.0)	3.9 (0.7)	>0.05
**Simulator C**			
Educational Value (SD)	3.4 (1.3)	1.3 (1.0)	>0.05
Fidelity (SD)	3.2 (0.8)	3.2 (0.9)	>0.05
Usability (SD)	3.8 (0.7)	4.0 (0.7)	>0.05
Overall Satisfaction (SD)	3.8 (1.0)	3.3 (0.9)	>0.05
**Simulator D**			
Educational Value (SD)	3.2 (1.1)	2.9 (1.0)	>0.05
Fidelity (SD)	2.6 (0.7)	3.0 (1.1)	>0.05
Usability (SD)	3.5 (0.5)	3.2 (1.5)	>0.05
Overall Satisfaction (SD)	3.2 (1.3)	3.0 (1.2)	>0.05

Means for both groups combined on the individual questionnaires. The values refer to the Likert’s scale of 1 (least favorable answer) to 5 (most favorable answer).

**Table 5 diagnostics-09-00050-t005:** Comparing Means for All Scores Combined.

Comparing Means Collective Score, Individual Questionnaires	Simulator A MedaPhor	Simulator B Schallware	Simulator C Simbionix	Simulator D ViMedix
**Educational Value**				
Benefits for novices	3.8 (1.2)	4.7 (0.5)	4.4 (0.7)	4.2 (0.9)
Benefits for intermediates	3.1 (1.5)	4.3 (1.0)	3.3 (1.6)	2.9 (1.5)
Benefits for advanced	2.1 (1.1)	2.7 (1.2)	2.2 (1.3)	2.0 (1.1)
**Fidelity**				
Scans resemble real life images	3.5 (1.2)	4.8 (0.4)	2.5 (1.2)	2.5 (1.2)
Probe resemble real life use	2.5 (1.5)	3.7 (1.1)	4.0 (1.2)	3.9 (1.0)
Knobs resemble real life use	3.1 (1.6)	3.5 (1.3)	3.2 (1.4)	2.1 (1.2)
**Usability**				
Intuitive use	3.7 (1.3)	4.4 (0.9)	4.5 (0.6)	4.2 (1.3)
Useful instructions	3.1 (1.2)	3.9 (1.1)	3.9 (0.9)	3.1 (1.4)
Useful feedback	3.1 (1.1)	3.7 (1.0)	3.3 (1.1)	2.7 (1.2)
**Overall Satisfaction**				
Easy to use	3.3 (1.5)	4.3 (0.9)	4.5 (0.6)	4.1 (1.5)
Overall feel	2.9 (1.6)	3.7 (1.0)	3.7 (1.2)	3.0 (1.5)
Recommend it to department	2.4 (1.4)	3.6 (1.1)	2.6 (1.5)	2.3 (1.3)

Means compared using collective scores for all participants from the individual simulator questionnaires. The values refer to the Likert’s scale of 1 (least favorable answer) to 5 (most favorable answer).

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
