# Peer review of "Four Virtual-Reality Simulators for Diagnostic Abdominal Ultrasound Training in Radiology"

_diagnostics, 2019, doi:10.3390/diagnostics9020050_

Round 1
Reviewer 1 Report
This study compared four different types of virtual-reality simulators for use in abdominal ultrasound training in radiology including 15 participants. As a result, the authors found all the four simulators were useable for diagnostic abdominal ultrasound and learning benefits were ranked higher for novices for more experienced ultrasonographers. The manuscript is well written; however, there some concerns described below.
My comments are as follows.
1. The number of participants for evaluating the four simulators was relatively small. I recommend the authors to mention this weak point in the discussion part.
2. In table 3, simulator B was most favorable in seven of the nine ranked areas; however, in table 4, the scores of the simulator B in terms of usability or overall satisfaction were relatively low. Please explain the discrepancy.
3. In this study, only two cases were evaluated in each simulator. I recommend the author to evaluate more cases.
Author Response
Thank you for your valuable comments. Please receive our responses below.
1. This has been added in the discussion:
“The limited number of participants also calls for care of interpretation and elaboration on the results.”
2. We sincerely thank you for noticing this error in table 4. We re-did the analysis and have updated the revised table with the correct results.
3. We agree, however the logistic parameters of the study did not allow for more than two cases, as the participants would have to use all simulators in one session for direct comparison. We therefore assessed time usage, general concentration spans and gain of additional knowledge with the result being the two-case setup.
Reviewer 2 Report
Dear Authors,
This manuscript is well-written and interesting in terms of raising the quality of diagnostic ultrasound training.
I have only few and minor suggestions for changes, shown in the document attached.
Comments and suggestions for changes are listed below. Suggestions for linguistic formulations or changes are highlighted in red. Suggestions for text to be deleted are marked with strikethrough.
Materials and Methods
1. Page 2, Table 1 Tabel 1:Demografics Table 1. Demographics
2. Page 2, Table 2 Tabel 1 The simulators Table 2. The simulators
3. Page 3, line 60
…with two cases: a A patient with…
Results
4. Page 3, line 82
…orange and red smileys, respectively
5. Page 4, Table 3
Tabel 3 Comparative QuestionnaireTable 3. Comparative Questionnaire
6. Page 5, line 111 Please delete a dot
…(most favourable answer). .
7. Page 5, Table 4 Tabel 4: Compare Mean by Groups Table 4. Comparing Mean Scores by Groups 8. Page 6, Table 5 Please complete the table heading if necessary Table 5: Comparing Means Table 5. Comparing Means …?...
9. Page 6, line 128
…and novices therefore could…
Discussion
10. Page 6, line 138 Not in present time?
… and ultrasound training had has a high‐ranking placement…
11. Page 7, line 157 Consider replacing the word “personal” with “personnel”
…and ensure that all trained personal personnel have…
12. Page 7, line 163 In Table 2, column 3, last row: Obstetrics/Gynecology… but in line 163 it [gynecology] is spelled with an “a”. Consider deleting the letter “a”.
…e.g. gynaecology…
Author Response
Thank you for your positive and valuable comments.
All suggested changes have been added to the manuscript.
Materials and Methods
1. Tabel 1:Demografics Table 1. Demographics
2. Tabel 1 The simulators Table 2. The simulators
3. …with two cases: a A patient with…
4. …orange and red smileys, respectively
5. Tabel 3 Comparative QuestionnaireTable 3. Comparative Questionnaire
6. …(most favourable answer). .
7. Tabel 4: Compare Mean by Groups Table 4. Comparing Mean Scores by Groups
8. Table 5: Comparing Means Table 5. Comparing Means …?...
9. …and novices therefore could…
10. … and ultrasound training had has a high‐ranking placement…
11. …and ensure that all trained personal personnel have…
12. …e.g. gynaecology…
Reviewer 3 Report
This article is interesting approach to use the real simulators for training the ultrasound technician and is also really helpful to give good discussion and guidance to select which simulator is good for the training. English grammar is very clear to be understood. The structure of the manuscript is well provided too. The references in the article are also recently published.Therefore, this article can be accepted with minor revision. However, authors had better change the following comments before final publications. 1. Please provide more clear Table 1. 2. In Figure 1, authors had better mark which A,B,C, and D are which company simulator even though Table 3 is already shown. Figure 1 needs a label.
Author Response
Thank you for your positive and valuable comments. All suggested changes have been added to the manuscript – please see the comments below.
1. The format has been changed.
2. This has been added to table 1.
3. The Figure has been labeled: “Simulators in the Study’s Setting”
Round 2
Reviewer 1 Report
The authors adequately responded to my comments.